# Guinea grass (*Megathyrsus maximus*) agronomic performances in mixed cultivation using irrigation condition, North-Western Ethiopia

**Mihret Anteneh Wolele, Fentahun Meheret Zeleke** *, **Yeshambel Mekuriaw Chekol, Bimrew Asmare Limenih, Zemenu Birhan Zegeye**

Department of Animal Sciences, College of Agriculture and Environmental Science, Bahir Dar University, Bahir Dar, Ethiopia

* fentahunmeheret@gmail.com

**Data Availability Statement:** All relevant data are within the manuscript and its Supporting Information files.

## Abstract

In this field experiment, the agronomic performance of Guinea grass was evaluated to assess the optimum planting spacing and harvesting age in mixed cultivation under irrigation conditions. These factors were arranged according to a 3*3 factorial setting, which was fitted to a randomized complete block design and replicated three times. This trial used three planting spaces (20, 30 and 40 cm) and harvesting ages (60, 90 and 120 days), grown in a mixed cropping system with Silver leaf desmodium using a furrow irrigation. The collected data includes plant height, leaves per plant, leaf length per plant, leaf to stem ratio, number of tillers per plant, number of roots per plant, root length per plant, and dry matter yield. The results of the study showed that age at harvest and planting space had a significant (P<0.001) effect on the morphological characteristics and dry matter yield of Guinea grass. The effect of these management practices was also affected the dry matter, crude protein and crude fiber contents of the grass. The maximum performances (high records) of the morphological characteristics and dry matter yield of Guinea grass were obtained by allocating a planting space of 40 cm and prolonging the harvesting period up to 120 days. However, harvesting at 90 days (11.48% of crude protein) resulted in the optimum nutritional contents. Therefore, in the area with irrigation facilities at midland agroecology, a planting space of 40 cm and a harvesting time of 90 days could be suggested through irrigation application in the study area and similar agro-ecologies.

## Introduction

Guinea grass (*Megathyrsus maximus*) is a member of the *Megathyrsus* genus and the Poaceae family. Tropical countries primarily use Guinea grass as a feed resource due to its high growth rate [1] and high seed and leaf production [2]. Under rainy and irrigated conditions, the annual dry matter yield per hectare was up to 12.2 tons/ha [3] and 42.7 tons/ha [4],

**Funding:** The author(s) received no specific funding for this work.

**Competing interests:** The authors declare no conflict of interest in this manuscript

respectively. It is suitable for cultivation using irrigation due to its tolerance to drought and saline soil conditions [5]. On the other hand, a feed shortage (quality and quantity) during the dry season strictly reduces livestock productivity in Ethiopia [6]. In this context, the livelihoods of smallholder farmers in terms of per capita consumption of animal products became low and far from the required level [7]. This could be improved by exploring potential forage resources through various means to alleviate sectorial constraints to sustain livestock production [6]. One of the interventions is to scale up forage production using potential grass species such as Guinea grass, which showed promising dry matter yield in a field trial at midland agro-ecology [8] but has not yet been adopted for optimum effect of harvest age and planting space. The irrigated areas of the country regularly used to grow cash crops, particularly in Mecha District, north-western Ethiopia. On the contrary, feed shortages have profoundly impacted dairy and beef farming in the district, where the number of farms has grown year after year due to the proximity of Bahir Dar, Amhara regional capital to the livestock market [9]. Therefore, the present study was conducted to figure out the effect of planting space and harvesting age on the performance of agro-morphological characteristics and nutritional contents of Guinea grass grown under mixed farming system with Silver Leaf Desmodium using irrigation in the midland agro-ecologies of Ethiopia.

## Materials and methods

### Description of the study area

The filed trial was located at Mecha district, Amhara regional state, north western Ethiopia. It is 520 kilometres from North West of Addis Ababa, capital city of Ethiopia. Geographically, it is located at a latitude of 11˚ 29' N, longitude 37˚ 29' E, and altitudes between 1807 and 2300 m above sea level (Fig 1). Annual precipitation ranged between 1240 and 1537 mm, with the temperature ranging from 28.01˚C to 10.57˚C. The dominant soil type is Nitosol. The common feed resources are natural pasture hay from pastureland, and agricultural and industrial by-products.

### Experimental design, layout, and treatments

A total of nine treatments (3*3) were arranged in a factorial arrangement using the variable planting space (PS) and age of harvest (AH) as presented in Table 1 below. Each treatment was replicated three times and planted in three blocks using a randomized complete block design (RCBD).

### Land preparation, planting materials and management

The experimental land was deliberately selected with irrigation infrastructure. The selected land ploughed three times. The land used for this trial was sized to 14m x 37m. It was prepared manually for the plantation. There were 27 plots and each sized 3m x 3m. The plots were separated by 1m and 1.5m strip to separate the blocks. The space between the plants was 50 cm. In February 2021, a total of 36 plants were planted using root-splitting within the plots and Silver leaf Desmodium *(Desmodium uncinatum (Jacq.) DC.)* was mixed at a rate of 25 g per row between the rows following the establishment of Guinea grass after two weeks [10]. The purpose of planting of this legume was the provision of ancillary services such as soil fertility improvement. All agronomic treatments, with the exclusion of fertilizer application were in place throughout the experimental period. It was irrigated three times per week during the period of seedling establishment and once a week after it was established.

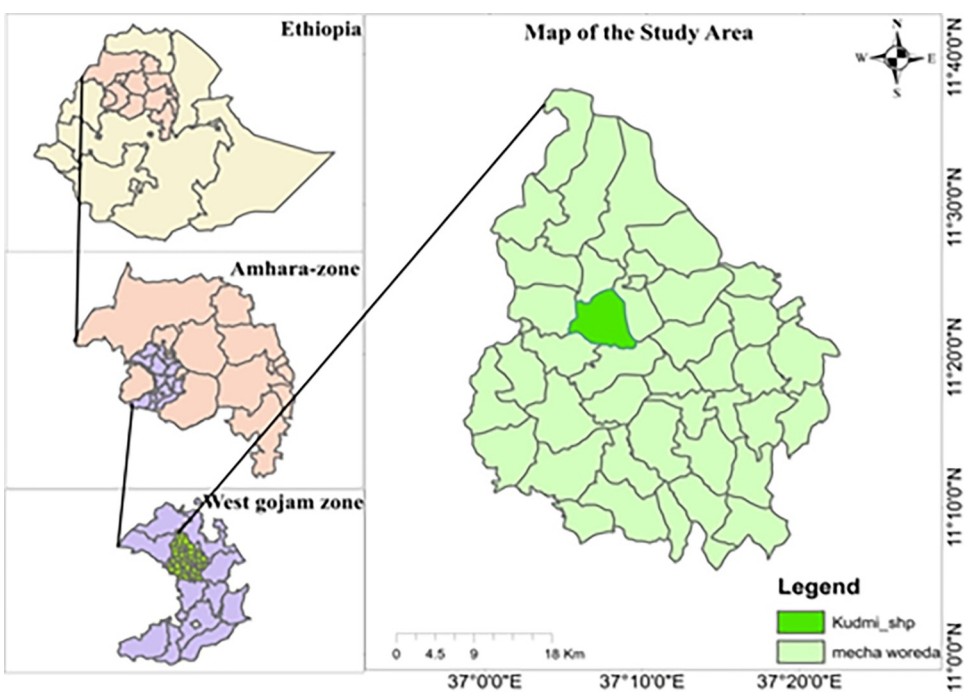

**Fig 1.**

## Data collection

**Plant morphological parameters and dry matter yield.** Morphological parameters were measured at 30 days interval using 10 plant samples grown in the middle row to avoid border effects. The biometric measurements used to describe Guinea grass were plant height (PH), leaf length per plant (LLPP), leaf to stem ratio (LR), number of leaf per plant (LNPP), number of tillers per plant (NTPP), number of roots per plant (NRPP) and root length per plant (RLPP). Total Fresh Weight (TFW) for Guinea grass was measured separately after harvest at 5cm above the ground using a 50 kg x 200 g spring balance scale inside a 1m x 1m area of the quadrant. A sub-sample of fresh weight (500 g) was transported to the laboratory in a plastic bag. The sample was dried for 24 hours at 105°C. The yield of dry matter (t/ha) of samples was calculated as follows [11].

$$\text{DMY t/ha} = \frac{TFW X DWss X 10}{HA X FWss}$$

Where:

**Table 1. List of treatment arrangements using planting space and age of harvest.**

| Treatment | Planting space(cm) | Age of harvesting (days) |
|---|---|---|
| 1 | 20 | 60 |
| 2 | 30 | 60 |
| 3 | 40 | 60 |
| 4 | 20 | 90 |
| 5 | 30 | 90 |
| 6 | 40 | 90 |
| 7 | 20 | 120 |
| 8 | 30 | 120 |
| 9 | 40 | 120 |

DMY = Dry matter yield.

TFW = total fresh weight kg/plot;

DWss = dry weight of sub-sample in grams;

FWss = fresh weight of sub-sample in grams,

HA = area of plot harvested in square meters and 10 is a constant for converting yields in $kg/m^2$ to t/ha.

**Chemical composition analysis.** In this trial, 1 mm sieved bulk samples (500g) were used for wet chemical analysis. The oven method of drying at 105°C for 24 hours was used to determine dry matter (DM). Total nitrogen (N) and ash were determined using the method described already in AOAC [12]. The neutral detergent fibre (NDF), acid detergent fibre (ADF), and acid detergent lignin (ADL) of the fibre components were analyzed using the defined producers [13].

**Statistical analysis.** Analysis of variance (ANOVA) for agro-morphological characteristics (S1 File) and chemical composition contents (S2 File) was executed using a linear model (lm) using the package 'agricolae' from R software v. 4.3.2 and the 'Metan' package for the analysis of Pearson correlations between the variables. The significance differences (P<0.05) between the individual mean values were separated using the Duncan Multiple Range Test [14].

The statistical model for analyzing the variances fitted with the following model:-

$$Yij = \mu + PSi + AHj + PSi*AHj + eij \tag{1}$$

Where:

$Y_{ij}$ = Agro- morphological characteristics and chemical composition performances responses;

$\mu$ = Overall mean;

$PS_{i = i}^{th}$ effect of planting space (20, 30 and 40 cm);

$HA_{j = j}^{th}$ effect of age of harvest (60, 90d and 120 days);

$Psi* AHj$ = interaction effect of planting space and harvesting age

$e_{ij}$ = random error

# Results

## Agro-morphological characteristics as a function of planting space and age of harvest

**Plant height.** The tallest plant height (75.89 cm) of Guinea grass was recorded at 40 cm planting space. It was also recorded at 120 days of harvest. The shortest plant height was recorded at 20 cm and the age of 60 days. It resulted due to the significant effect (P<0.001) of planting space and age of harvest on plant height performances. In this study, the age at harvest had a greater effect on the height of the plant than the planting space (Table 2).

**Number of tillers per plant.** In this field trial, Guinea grass had more tillers per plant (52.08) when counted at 120 days. It had the lowest number of tillers when counted at 60 days. At a significant level of P<0.001 and P<0.05, Guinea grass tillers per plant were more influenced by the number of harvesting days than planting space respectively.

**Number and length of leaf per plant.** The longest harvest (120 days) of Guinea grass produced high leaf counts (401.90) and length per plant (27.12cm). It had the lowest leaf number (253.52) and length (18.86 cm) due to its early harvest age and used the narrow planting space.

**Number of root and root length per plant.** The number of root per plant and its length became highest at 120 days and lowest at 60 days. The 40 cm planting space had a higher root number per plant than the 20 cm and 30 cm planting spaces. Guinea grass root length required an elongated harvest age (high score) rather than planting space allocation (low score).

**Table 2. Effect of planting space and harvesting age on morphological characteristics and dry matter yield intercropped Guinea grass with sliver leaf desmodium.**

| Parameters | PH/cm | LNPP | NLPP | LLPP/cm | NRPP | RLPP/cm |
|---|---|---|---|---|---|---|
| | | | Planting space | | | |
| 20 | 66.33[b] | 43.03[b] | 333.64[b] | 22.34[b] | 90.58[c] | 6.17[ab] |
| 30 | 73.78[a] | 43.70[ab] | 340.23[a] | 22.81[b] | 92.80[b] | 6.41[ab] |
| 40 | 75.89[a] | 45.10[a] | 346.02[a] | 23.98[a] | 94.96[a] | 6.63[a] |
| P-value | <0.001 | <0.05 | <0.01 | <0.01 | <0.001 | <0.05 |
| | | | Age of harvest | | | |
| 60 | 57.89[c] | 36.51[c] | 253.52[c] | 18.86[c] | 76.27[c] | 5.66[c] |
| 90 | 71.78[b] | 43.24[b] | 364.48[b] | 22.76[b] | 96.24[b] | 6.37[b] |
| 120 | 86.33[a] | 52.08[a] | 401.90[a] | 27.12[a] | 105.83[a] | 7.18[a] |
| Overall mean | 72 | 43.94 | 339.96 | 23.07 | 92.8 | 6.4 |
| MSE | 10.52 | 2.42 | 34.9 | 0.71 | 2.12 | 0.71 |
| CV | 4.51 | 3.54 | 1.74 | 3.66 | 1.57 | 13.2 |
| P-value | <0.001 | <0.001 | <0.001 | <0.001 | <0.001 | <0.001 |

[abc] Superscript showed means significant difference of variables in column at P < 0.01, P < 0.001 and P>0.05; Plant height (PH), tillers number per plant (LNPP), number of leaf per plant (NLPP), leaf length per plant (LLPP), number of roots per plant (NRPP); root length per plant (RLPP), mean square error (MSE), and coefficient of variation (CV).

## Interaction effect of planting space and age of harvest

**Leaf to stem ratio.** Leaf to stem ratio was significantly (P<0.01) more affected by planting space and age of harvest. The highest (1.74) records were obtained at wider planting space (40cm) and early harvest age (60 days) (see Table 3). The lowest (about 1.3) records were obtained at the 120-day harvest at planting spaces of 20cm, 30cm and 40cm.

**Dry matter yield.** The dry matter yield (ton/ha) of guinea grass grown under an irrigation system was highly (P< 0.001) influenced by the interaction effect of both planting space and harvest age as leaf to stem ratio. In this study, the highest dry matter yield was obtained at 40

**Table 3. The interaction effect of planting space and age of harvest over the performance of leaf to stem and dry matter yield.**

| Planting space (PS) | Age of harvest (AH) | Leaf to stem ratio | Dry matter yields ton/ha |
|---|---|---|---|
| 20cm | 60 days | 1.45[b] | 8.97[g] |
| | 90 days | 1.39[b] | 10.33[ef] |
| | 120 days | 1.26[c] | 12.06[bcd] |
| 30cm | 60 days | 1.67[a] | 10.11[f] |
| | 90 days | 1.42[b] | 11.37[cde] |
| | 120 days | 1.27[c] | 12.64[bc] |
| 40cm | 60 days | 1.74[a] | 11.08[def] |
| | 90 days | 1.4[b] | 12.06[b] |
| | 120 days | 1.28[c] | 14.26[a] |
| Overall mean | | 1.44 | 11.47 |
| MSE | | 0.003 | 0.42 |
| CV | | 3.94 | 5.63 |
| P-values | | <0.01 | <0.001 |

[abcdefg] superscript showed a level of mean differences of variables across the columns at P < 0.01 and P < 0.001, mean square error (MSE), coefficient of variation (CV).

cm planting space and 120 days harvesting age. The lowest (8.97ton/ha) dry matter yield was harvested in combined management of 20 cm planting space and 60 days harvesting age (see Table 3).

## Effect of plant spacing and harvesting age on the chemical composition of Guinea grass

Guinea grass chemical composition was highly (p<0.00) influenced by planting space with maximum values of 92.26% at 40cm as shown in Table 4. At 20 cm planting space it was low (89.12%). The chemical content of this grass grows significantly (p<0.001) different from the age of the harvest when irrigated in midland agroecology. Similarly, the percent of organic matter (OM %) content was highly influenced individually by planting space and harvest age with P values of P<0.001 (Table 4). The maximum percept of OM was contributed by 120 days of harvest. The early age of harvest (60 days) and wider space resulted in a low amount of OM contents.

The crude protein (CP) content of irrigated guinea grass was significantly affected by harvest ages (p<0.001), followed by planting space (p<0.05). The high concentration of CP which was 12.03% and 12.70% records were obtained from wider space (40 cm) and at the early age of harvest (60 days) respectively. The CP content of 10.48% was the lowest record obtained at the prolonged age of harvest. In addition, crude fiber content with NDF, ADF and ADL contents was strongly influenced by harvest age (P< 0.001), followed by planting space with P<0.05, P<0.001 and P<0.01 for the performances of NDF, ADF and ADL contents respectively. The NDF content became high (76.93%) at 120 days and low (65.93%) at 60 days. It was also high at 20 cm and low at 40 cm with scores of 73.79% and 68.61% respectively. The ADL content followed as pattern of NDF content. However, ADF was high at 40cm which accounted for 73.79%.

## Correlation among morphological characteristics, dry matter yield and chemical composition of Guinea grass

The Pearson's correlation (Metan package) analysis showed that Guinea grass plant height showed a significant (P<0.001) positive correlation with NRPP, RLPP and NLPP (Fig 2). The LSR of this grass showed a significant negative correlation (p<0.001) with the morphology of

**Table 4. Effect of planting space and harvesting age on the chemical composition of the Guinea grass.**

| Parameters | DM% | OM% | CP % | NDF% | ADF% | ADL% |
|---|---|---|---|---|---|---|
| | | | Planting space(cm) | | | |
| 20 | 89.12[c] | 85.78[a] | 11.15[b] | 73.79[a] | 43.18[b] | 8.02[a] |
| 30 | 91.25[b] | 84.97[a] | 11.47[ab] | 72.35[a] | 43.66[b] | 7.79[a] |
| 40 | 92.26[a] | 85.87[b] | 12.03[a] | 68.61[b] | 45.18[a] | 6.86[b] |
| P-values | <0.001 | <0.001 | <0.05 | <0.05 | <0.001 | <0.01 |
| | | | Age of harvest(cm) | | | |
| 60 | 89.66[c] | 83.80[c] | 12.70[a] | 65.93[c] | 42.98[c] | 6.15[c] |
| 90 | 90.85[b] | 84.72[b] | 11.48[b] | 71.99[b] | 44.11[b] | 7.24[b] |
| 120 | 92.11[a] | 86.12[a] | 10.48[c] | 76.93[a] | 45.15[a] | 9.28[a] |
| Overall mean | 90.87 | 84.87 | 11.55 | 71.58 | 44.08 | 7.55 |
| MSE | 0.64 | 0.7 | 0.43 | 12.93 | 0.65 | 0.38 |
| CV | 0.88 | 0.98 | 5.64 | 5.02 | 1.83 | 8.13 |
| P-values | <0.001 | <0.001 | <0.001 | <0.001 | <0.001 | <0.001 |

[abc] Mean with different superscripts in a row were significantly different at P < 0.001, Dry matter (DM), Organic matter (OM), Crude protein (CP), Neutral detergent fibre (NDF), Acid detergent fibre (ADF), Acid detergent lignin (ADL), mean square error (MSE), and coefficient of variation(CV)

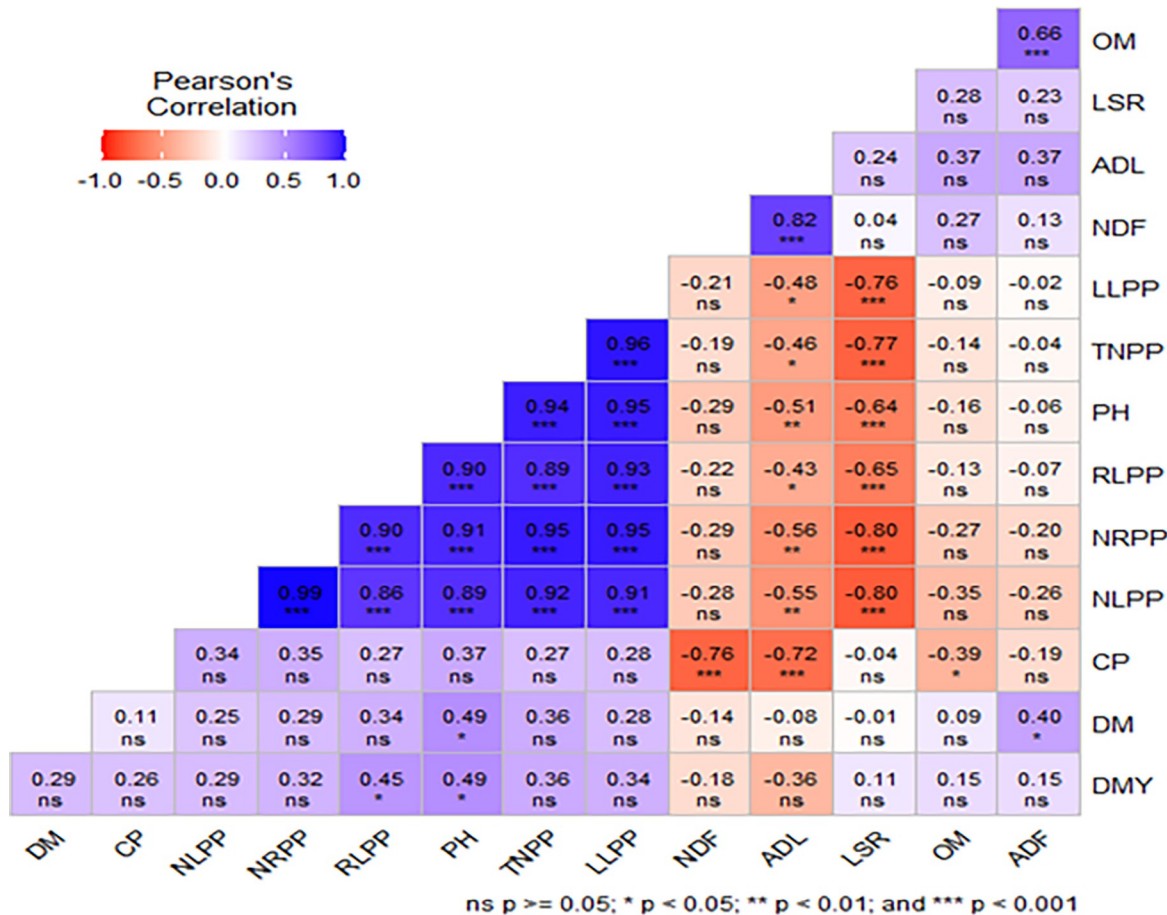

**Fig 2.**

the plants in NLPP, NRPP, PH, TNPP and LLPP. There was a negative correlation between NLPP and LSR, but a strong positive correlation between TNPP, PH, RLPP and NRPP. There was a strong positive correlation between LLPP and NLPP, NRPP, RLPP, PH and TNPP. Furthermore, there was a negative correlation and a significant negative correlation between the CP content of guinea grass and NDF, ADL and OM.

## Discussion

### Morphological characteristics performance of Guinea grass

Above-ground morphometric performances of guinea grass such as plant height, number of tillers, number of leaves and leaf length per plant reached their maxima by increasing the planting space from 20cm to 40cm. It brings significantly high performance on morphologic changes in comparison of plants from the shortest plant to the tallest plant. It had a maximum score of the number of tillers, number of leaf and length per plant. Such high morphological performance could benefit smallholder farmers to subsidise feed shortage feed in the dry season. It is similar to the current study showed a high score of morphological characteristics obtained as the age of harvest increases by up to 120 days. This is worthy to have adequate feed for livestock production. The consequence of this experiment in coincidence with the prescription of the optimal planting space and the delay of the harvesting age improved the productivity of the grass species

[15,16]. Because, the performance of morphological characteristics of Guinea grass was improved to the tallest, largest number of tiller and leaf per plant in this trial when it was managed at wider space and elongated age of harvest. It might be related to adequate nutrients and moisture availability in a wider space [17]. It is similar to wider plating space contributing more tiller number than the narrow planting space as reported for Napier grass [18], Desho grass [19] and Mulat grass [20]. Similarly, age of harvest elongation had a response for morphological characteristic advancement for tropical grass species like Napier grass [21], Desho grass [19] and Para grass [22]. On the other hand, in this study, root morphological performance improved in parallel with increasing planting space and delayed harvest age, but root length was more affected by harvest age than planting space. It is in line with previous reports when the score of roots was done close to 120 days as reported for Desho grass [19] and Para grass [22].

### Dry matter yield and leaf to stem ratio

The amount of dry matter (DMY) produced per hectare as well as the estimated leaf to stem ratio fluctuated greatly (P<0.001) depending on the plating space and the age of harvest for the irrigated Guinea grass. Dry matter yield increased significantly at 40 cm and 120 days, while LSR was maximal at 60 days. At a late harvest age (120 days), high dry matter was obtained and minimal at a harvest of 60 days. This was comparable to the dry matter yield (12.72 t/ha) obtained from the Desho grass grown with the vetch species [23]. Moreover, in this study, the 40 cm planting space produced a higher dry matter yield (14.26 t/ha) than the 50 cm planting space (5.41 t/ha) employed during mixed pasture cultivation under rain fed conditions [24]. As a result, due to the advancement of plant morphology, in the current field experiment, the dry matter yield of Guinea grass was strongly influenced by plant spacing and harvesting age. It is in line with the finding when the Chicory plant intercropped with dwarf Elephant grass (4.659 kg/ha DM); established at plant spacing of 25x 25cm and harvested at 60 days [25], mono cultivation of Columbus grass (18.83 t/ha of leaf DM) planted at 25×35cm plant spacing and harvested at 8weeks [26] and Mulato cultivars (12.34.t/ha DM) grown at 30x50cm and harvested at 120 days under irrigation conditions [27]. The dry matter yield differences recorded for Guinea grass might be due to agroecology, the genetic makeup of the species and management aspects. On the other hand, The 40cm planting space produced a maximum LSR score of 1.74 after 60 days of harvest, which was superior to 1.69 from legume combination cultivations of Desho grass harvested after 90 days [23].

### Effect of planting space and age of harvest on the chemical composition of Guinea grass

In the current study, the protein content of Guinea grass grown with Silver leaf desmodium was significantly different as a function of planting space and harvest age. It was higher when planted in a 30cm space and harvested after 90 days. It is explained by the fact that the *A. gayanus* and *Stylosanthes* mixture had the maximum protein concentration (152 g/kg), which is an indicator of quality animal feed produced from the intercropping of legumes and grass species [28]. It is also consistent with the nutrient content of intercropped species of grasses and legumes, which maximizes the nutritional value of pasture in lowland agroecology [29].

### Correlation among morphological characteristics, dry matter yields and chemical composition of Guinea grass

The positive correlation of most morphological characteristics and dry matter yield of Guinea grass in this experiment indicated that optimum planting space and harvesting age could

enhance the morphological and dry matter yield performance of grass species like the productivity from natural pasture when intercropped with legumes [29] It is similar results to the dry matter yield performance of desho grass associated with morphological parameters [19]. On the other hand, the crude protein content was positively correlated with leaf to stem ratio and crude fiber contents (NDF, ADF and ADL) moderately positive association with most morphological characteristics. It confirms that the intercropping system increased the nutritive values of feed resources of grass species [23,29].

## Conclusion

In the current experiment, variable planting space had a strong effect on the morphological characteristics, dry matter yield and nutrient content of guinea grass in mixed cropping. It had a significant role in the achievement of maximum dry matter yields and showed better development of plant morphology. It is uniform for most morphology characteristics such as PH, NTPP, NLPP, LL, NRPP, RLPP and DMY, which are higher at late harvest age (120 days). It is also reflected in the nutrient contents (DM, Ash, OM, ADF and ADL) of the grass, except for CP which was high at an early age (60 days). Consequently, this experiment concludes that for the nutrient and dry matter production of Guinea grass, a wider plant spacing of 40 cm and an intermediate harvest age of 90 days are both more satisfactory. It could be an alternative fodder resource following further study under rain fed conditions. Moreover, animal evaluation trial and economic feasibility of forage production are of worth to be done as future research.

## Supporting information

**S1 File. Raw data collected from plant morphology and dry matter yield per ha.**
(PDF)

**S2 File. Raw data on chemical composition analysis of Guinea grass.**
(PDF)

## Acknowledgments

This experiment was accompanied by the support of Bahir Dar University, Bahir Dar and Jabi Tehinan District Agriculture Office and Sirinka Research Centre were provided plant material, Ethiopia.

## Author Contributions

**Conceptualization:** Mihret Anteneh Wolele.

**Data curation:** Mihret Anteneh Wolele.

**Methodology:** Mihret Anteneh Wolele.

**Supervision:** Fentahun Meheret Zeleke.

**Writing – original draft:** Fentahun Meheret Zeleke, Zemenu Birhan Zegeye.

**Writing – review & editing:** Yeshambel Mekuriaw Chekol, Bimrew Asmare Limenih.

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
