## [Decision Letter · Decision Letter 0]

26 Aug 2024

PONE-D-24-32761Guinea Grass (Panicum Maximum) Agronomic Performances in Mixed Cultivation using Irrigation conditions, Northern Western EthiopiaPLOS ONE

Dear Dr. Zeleke,

Thank you for submitting your manuscript to PLOS ONE. After careful consideration, we feel that it has merit but does not fully meet PLOS ONE’s publication criteria as it currently stands. Therefore, we invite you to submit a revised version of the manuscript that addresses the points raised during the review process.

We look forward to receiving your revised manuscript.

Kind regards,

Mohammad Nur Alam, PhD

Academic Editor

PLOS ONE

Journal requirements: 1. When submitting your revision, we need you to address these additional requirements. Please ensure that your manuscript meets PLOS ONE's style requirements, including those for file naming. The PLOS ONE style templates can be found at https://journals.plos.org/plosone/s/file?id=wjVg/PLOSOne_formatting_sample_main_body.pdf and https://journals.plos.org/plosone/s/file?id=ba62/PLOSOne_formatting_sample_title_authors_affiliations.pdf. 2. In your Methods section, please provide additional information regarding the permits you obtained for the work. Please ensure you have included the full name of the authority that approved the field site access and, if no permits were required, a brief statement explaining why. 3. In the online submission form, you indicated that [Datasets are available at the hand of corresponding author]. All PLOS journals now require all data underlying the findings described in their manuscript to be freely available to other researchers, either 1. In a public repository, 2. Within the manuscript itself, or 3. Uploaded as supplementary information.This policy applies to all data except where public deposition would breach compliance with the protocol approved by your research ethics board. If your data cannot be made publicly available for ethical or legal reasons (e.g., public availability would compromise patient privacy), please explain your reasons on resubmission and your exemption request will be escalated for approval. 

Additional Editor Comments:

Dear Authors

Thanks for your nice work. There are many errors in the MS. Be serious and follow the reviewers' comments, and strictly perform major revisions.

Regards

Mohammad Nur Alam, PhD

Reviewers' comments:

Reviewer's Responses to Questions

**Comments to the Author**

1. Is the manuscript technically sound, and do the data support the conclusions?

Reviewer #1: Partly

Reviewer #2: Yes

2. Has the statistical analysis been performed appropriately and rigorously? 

Reviewer #1: No

Reviewer #2: I Don't Know

3. Have the authors made all data underlying the findings in their manuscript fully available?

Reviewer #1: Yes

Reviewer #2: Yes

4. Is the manuscript presented in an intelligible fashion and written in standard English?

Reviewer #1: Yes

Reviewer #2: Yes

5. Review Comments to the Author

Reviewer #1: The authors have conducted good research work however; the MS needs major improvement before it can consider for publication. I have provided below some shortcomings identified in the MS and they need to be addressed.

General comments

1. In the abstract there must be research objectives, research methods, results and discussion, and the originality of the research.

2. The introduction should be concise and clear. Other than that, in quoting, you must use a new language and not copy and paste.

3. The research method must be clear about the research location, research methods that are in accordance with the research theory.

4. The results and discussion must be detailed, clear, concise, concise, and easy to understand.

5. Conclusions must be short and clear, provide research suggestions for further researchers

6. Must follow the format that is in accordance with the submitted journal.

Specific comments

1. Change the botanical name of guinea grass from Panicum maximum to Megathyrsus maximus.

2. Use a similar format for keywords.

3. Confirm the statistical design—whether it is CRBD (Completely Randomized Block Design) or RCBD (Randomized Complete Block Design).

4. Table 4: CP values are on the slightly higher side. Please check.

5. Include the soil fertility status of the experimental field.

Reviewer #2: Thank you for giving me the opportunity to review your manuscript entitled "Guinea grass (Panicum Maximum) Agronomic Performance for Mixed Cultivation using Irrigation conditions, Northern Western Ethiopia". I very much enjoyed reading your manuscript. The authors purpose that at 40 cm plant spacing with 90 day's harvest age is better for nutritional value of Guinea grass however maximum other parameters are given better result with 120 day's harvest age.

The manuscript is well written. It's may be typing mistake in 43 & 281 which is Sliver Leaf Desmodium actually it Silver Leaf Desmodium.

Discussions are logical with tables and figures.

References are suitable for this manuscript, I think if 8,10,13,16,17,19,& 23 references are same style as like 2 it will be better.

It is a nice work I wish it will help the growers to fulfill their cattle feed.

6. PLOS authors have the option to publish the peer review history of their article (what does this mean?). If published, this will include your full peer review and any attached files.

Reviewer #1: No

Reviewer #2: No

---

## [Author Response · Author response to Decision Letter 0]

23 Sep 2024

Dear Editor,

I would like to thank you for giving us feedback to improve our MS through your committed work and the anonymous reviewers’ efforts. 

We have gone through your comments and revised accordingly and prepared an author’s response (submited). We also well come further comments to qualify the manuscript.

---

## [Decision Letter · Decision Letter 1]

25 Nov 2024

PONE-D-24-32761R1Guinea grass (Megathyrsus maximus) agronomic performances in mixed cultivation using irrigation condition, North-Western EthiopiaPLOS ONE

Dear Dr. Zeleke,

Thank you for submitting your manuscript to PLOS ONE. After careful consideration, we feel that it has merit but does not fully meet PLOS ONE’s publication criteria as it currently stands. Therefore, we invite you to submit a revised version of the manuscript that addresses the points raised during the review process.

**ACADEMIC EDITOR: **Please see the attached word file. ==============================

We look forward to receiving your revised manuscript.

Kind regards,

Prabhu Govindasamy, Ph.D.

Academic Editor

PLOS ONE

Journal Requirements:

Additional Editor Comments:

There are a few minor suggestions and comments I have made for authors. Kindly resubmit it.

Reviewers' comments:

Reviewer's Responses to Questions

**Comments to the Author**

1. If the authors have adequately addressed your comments raised in a previous round of review and you feel that this manuscript is now acceptable for publication, you may indicate that here to bypass the “Comments to the Author” section, enter your conflict of interest statement in the “Confidential to Editor” section, and submit your "Accept" recommendation.

Reviewer #1: All comments have been addressed

Reviewer #2: All comments have been addressed

2. Is the manuscript technically sound, and do the data support the conclusions?

Reviewer #1: Yes

Reviewer #2: Partly

3. Has the statistical analysis been performed appropriately and rigorously? 

Reviewer #1: Yes

Reviewer #2: I Don't Know

4. Have the authors made all data underlying the findings in their manuscript fully available?

Reviewer #1: Yes

Reviewer #2: Yes

5. Is the manuscript presented in an intelligible fashion and written in standard English?

Reviewer #1: Yes

Reviewer #2: Yes

6. Review Comments to the Author

Reviewer #1: I thank the authors to largely address my comments. Therefore, I think the MS can be accepted in its present form.

Reviewer #2: Guinea grass (Megathyrsus maximus) agronomic performances in mixed cultivation using irrigation condition, North-Western Ethiopia is a good work that will help to increase the cultivation of animals fodder. All comments have been addressed by the authors carefully.

7. PLOS authors have the option to publish the peer review history of their article (what does this mean?). If published, this will include your full peer review and any attached files.

Reviewer #1: No

Reviewer #2: No

---

## [Author Response · Author response to Decision Letter 1]

9 Dec 2024

Dear Editor,

I would like to thank the academic editor of PLOS ONE and the anonymous reviewers for providing us with suggestions that assisted us in improving the content of our paper. We have incorporated your comments, amended them, and prepared the response shown in the table below. We are happy to receive further comments, thank you in advance.

---

## [Editor Report · Decision Letter 2]

13 Dec 2024

Guinea grass (Megathyrsus maximus) agronomic performances in mixed cultivation using irrigation condition, North-Western Ethiopia

PONE-D-24-32761R2

Dear Dr. Zeleke,

We’re pleased to inform you that your manuscript has been judged scientifically suitable for publication and will be formally accepted for publication once it meets all outstanding technical requirements.

Kind regards,

Prabhu Govindasamy, Ph.D.

Academic Editor

PLOS ONE

Additional Editor Comments (optional):

Thank you for attending all the comments and suggestions made by reviewers and editor. 

Reviewers' comments:

<gdiv id="ginger-floatingG-container" style="position: absolute; top: 0px; left: 0px;"><gdiv class="ginger-floatingG ginger-floatingG-closed ginger-floatingG-posdown ginger-floatingG-spin ginger-floatingG-dirty" style="display: block; left: 650.1px; top: 158.8px; z-index: 51;"><gdiv class="ginger-floatingG-disabled-main"><gdiv class="ginger-floatingG-bar-tool-tooltip ginger-floatingG-bar-tool-tooltip-enable">Enable Ginger</gdiv></gdiv><gdiv class="ginger-floatingG-offline-main"><gdiv class="ginger-floatingG-bar-tool-tooltip">*Cannot connect to Ginger* Check your internet connection

or reload the browser</gdiv></gdiv><gdiv class="ginger-floatingG-enabled-main"><gdiv class="ginger-floatingG-bar"><gdiv class="ginger-floatingG-bar-tool ginger-floatingG-bar-tool-disable"><ga></ga><gdiv class="ginger-floatingG-bar-tool-tooltip">Disable Ginger</gdiv></gdiv><gdiv class="ginger-floatingG-bar-tool ginger-floatingG-bar-tool-rephrase ginger-floatingG-bar-tool-rephrase_big-circle"><ga class="ginger-floatingG-bar-tool-rephrase__btn" id="ginger__floatingG-bar-tool-rephrase__btn">Rephrase</ga><gdiv class="ginger-floatingG-bar-tool-tooltip ginger-floatingG-bar-tool-tooltip_rephrase">Rephrase with Ginger (Ctrl+Alt+E)</gdiv></gdiv><gdiv class="ginger-floatingG-bar-tool ginger-floatingG-bar-tool-mistakes"><ga>1</ga><gdiv class="ginger-floatingG-bar-tool-tooltip">Log in to edit with Ginger</gdiv></gdiv></gdiv></gdiv><gdiv class="ginger-floatingG__loading-popup">Ginger is checking your text for mistakes...</gdiv><gdiv class="ginger-floatingG__disabling-popup " style="display: none;">Disable Ginger in this text fieldDisable Ginger on this website</gdiv><gdiv class="ginger-floatingG-contentPopup" style="display: none;"><gdiv class="ginger-floatingG-contentPopup-wrap-limit">600/2139 free characters checked.Go Premium to check longer texts and entire documents</gdiv></gdiv></gdiv></gdiv>

---

## [Editor Report · Acceptance letter]

20 Dec 2024

PONE-D-24-32761R2 

PLOS ONE

Dear Dr. Zeleke, 

I'm pleased to inform you that your manuscript has been deemed suitable for publication in PLOS ONE. Congratulations! Your manuscript is now being handed over to our production team.

Kind regards, 

on behalf of

Dr. Prabhu Govindasamy 

Academic Editor

PLOS ONE